# Shrub Diversity and Niche Characteristics in the Initial Stage of Reconstruction of Low-Efficiency *Cupressus funebris* Stands

**Yang Wang** [1,2,3], **Silu Chen** [4], **Wenchun He** [1,2,3], **Jingjing Ren** [1,2,3], **Xiaochen Wen** [1,2,3], **Yu Wang** [1,2,3], **Xianwei Li** [1,2,3], **Gang Chen** [1,2,3], **Maosong Feng** [1,2,3] and **Chuan Fan** [1,2,3,*]

1    College of Forest, Sichuan Agricultural University, Chengdu 611130, China; wangyang@stu.sicau.edu.cn (Y.W.); 2019104001@stu.sicau.edu.cn (W.H.); Mj568791726@163.com (J.R.); 2020204010@stu.sicau.edu.cn (X.W.); 2020304062@stu.sicua.edu.cn (Y.W.); lxw@sicau.edu.cn (X.L.); g.chen@sicau.edu.cn (G.C.); 12352@sicau.edu.cn (M.F.)
2    Sichuan Province Key Laboratory of Ecological Forestry Engineering on the Upper Reaches of the Yangtze River, Chengdu 611130, China
3    State Forestry and Grassland Administration Key Laboratory of Forest Resources Conservation and Ecological Safety on the Upper Reaches of the Yangtze River, Chengdu 611130, China
4    Sichuan Forestry and Grassland Investigation and Planning Institute, Chengdu 610081, China; sal358501916@163.com
*    Correspondence: fanchuan@sicau.edu.cn

**Abstract:** The upper reaches of the Yangtze River are a very important ecological barrier in China, but the ecological benefits of large-scale *Cupressus funebris* Endl.plantations are low. This study investigated 12 plantations of different compositions and densities, including two densities of *Cinnamomum septentrionale* Hand.-Mazz. (Cs), *Alnus cremastogyne* Burk. (Ac), and *Toona sinensis* (A. Juss.) Roem. (Ts), and mixed plantations of Cs + Ac (CA), Ts + Cs (TC), Ts + Ac (TA), and Ac + Ts + Cs (ATC) and the cutting-blank (CB), and, at the same time, the unreconstructed pure *C. funebris* (Cf) forest was set as the control. We aimed to explore the influence mechanism of upper tree composition and density on shrub diversity, as well as the relationship between shrub diversity and niche. Our research results are as follows: (1) Among all the patterns, the TA, CA, and TC patterns are the most conducive to improving the diversity of shrubs. The composition and density of different trees have a great influence on the diversity of shrubs. (2) Niche is closely related to the diversity of shrubs. In the patterns of low niche overlap between dominant shrubs, the diversity of shrubs is greater. These results contribute to a deeper understanding of the relationship between the diversity of overstory and shrubs, and reveals the relationship between niche and diversity.

**Keywords:** *Cupressus funebris*; low-function plantation; mixed forest; plant diversity; niche; reconstruction

## 1. Introduction

The maintenance of biodiversity is a key issue in current forest management, and can assist human societies to guarantee the long-term persistence of forest ecosystems, services, and resources [1,2]. Understory shrubs, as backup plants for the regeneration, development, and succession of forest ecosystems, are the earliest woody plants in the process of succession and are able to drive the appearance and succession of the canopy [1]. The diversity of shrubs is an important component of forest biodiversity, especially in plantations, where is critical to the development of communities and the maintenance of diversity [2,3]. In addition, shrubs maintain the soil quality of the stand, and play an important role in forest development, the conversion of cropland to forests, etc. However, previous research on the diversity of plants has mostly focused on the diversity of trees, the diversity of grasses, or the trees and shrubs as a whole, not solely focusing on the study of the diversity of shrubs.

Investigation of the diversity differences and the niche characteristics of dominant shrub species in different forest stands is of great importance to revealing the position and the role of each shrub species and predicting the shrub diversity in the future community and the interrelationship among species. The diversity of a shrub is closely related to its niche. Additionally, the niche theory has made a great contribution to the formation and maintenance of the diversity of plants. For example, the basis of the biodiversity–environmental heterogeneity theory is that changes in the environment will lead to increased environmental heterogeneity, which increases the niche space within the forest stand, and improves the diversity of plants in forestland [4–6]. In addition, the higher the diversity of a plant is, the more niche space species can create, leading to higher environmental heterogeneity and an increase in species richness at the community scale [7,8]. The wider the average niche breadth is, the higher the competition among species and the greater the possibility of niche overlap, which causes lower diversity and productivity [9]. Niche differentiation is widely accepted as the main driving force for the coexistence of species, thus favoring the maintenance of biodiversity [10,11].

There are many biotic and abiotic conditions that influence the diversity of shrubs, such as the forest canopy composition [12,13], altitude and latitude [14], and the physical and chemical properties of soil [2]. They directly or indirectly affect the forest environments (such as the availability and quality of undergrowth sunlight, soil water, and nutrient availability), which are critical to forming and maintaining the diversity of shrubs. In particular, there is a complicated relationship between the canopy and the shrub [15]. This is due to the fact that the canopy controls the environmental resources that are important to the shrub layer, such as light and forest water [1,16]. Therefore, the composition structure of the canopy will also play an important role in the distribution and diversity of shrubs [2,17]. Some studies have indicated that the herb and shrub layer composition and diversity are not significantly affected by the species of overstory trees [18,19], but others have stated that the diversity and composition of the understory plant species depend on the density and composition of the overstory tree species [16,20,21].

However, for shrubs alone, the response to overstory composition and density and the response mechanisms are not clear. Further, most of the existing studies on plant diversity only discuss the various driving factors or evaluate the economic or ecological benefits of the diversity of different plant stands [22]. Few studies have analyzed the diversity of shrubs from the perspective of their niche. Thus, whether and how the composition and density of the upper layer of trees affects shrub diversity, and how shrub diversity and niches are linked are questions that remain to be addressed.

*Cupressus funebris* Endl., a subtropical evergreen coniferous tree, is the main vegetation restoration tree species in the upper reaches of the Yangtze River in China. Due to the high initial planting density and the monospecificity of *C. funebris*, the ecological status under the forest is not ideal. Therefore, it is necessary to reconstruct the *C. funebris* plantation and study the diversity of shrubs and the relationship between diversity and the niche after the reconstruction. In this study, we selected three broad-leaved tree species, *Cinnamomum septentrionale* Hand.-Mazz (Cs), *Alnus cremastogyne* Burk. (Ac), and *Toona sinensis* (A. Juss.) Roem. (Ts), with different densities, 1.5*2 m and 1.5*4 m, and compositions, Cs + Ac(CA), Cs + Ts(CT), and Ac + Ts(AT), to establish 11 different forest stands in this study, according to the principle of adapting trees to the site. Taking the pure *C. funebris* forest as the control, we attempted to study the diversity of shrubs in terms of the composition and density of the overstory. We also attempted to understand the relationship between the shrub niche and the diversity of shrubs in different model plantations by studying the niches of shrubs in different model plantations. The purpose is to provide information that improves our understanding of the influence mechanism of the composition and density of the overstory of the plantation on shrub diversity and its relationship with the niche, and help maximize the ecological benefits of shrubs in the plantation community.

This study aimed to: (a) discuss the impact of tree density and composition on shrub diversity, and possible impact mechanisms; and (b) by studying shrub niches in different

patterns of plantations, understand the relationship between shrub niches and shrub diversity in different patterns of plantations.

## 2. Materials and Methods

### 2.1. Study Area

The study area was located in the town of Yongxin (104°–104°32′ E, 31°05′–31°20′ N), Jingyang District, Deyang City, Sichuan Province, China (Figure 1). The soil is a purple soil. It is situated within the subtropical zone and is characterized as a humid monsoon climate. The altitude is 457–764 m, the mean annual temperature is 16.1 °C, the annual precipitation is 893.4 mm, and the mean annual sunshine is 1251.4 h, with 271 frost-free days per year on average. Before the reconstruction, the vegetation under the 30-year-old forest was sparse, and the quality of the habitat was poor. The main trees were *C. funebris*. The mean DBH, height, and canopy closure of *C. funebris* forest were 7 cm, 8 m, and 0.8, respectively. The shrubs under the forest stand were mainly *Broussonetia papyrifera* (Linnaeus) L'Heritier ex Ventenat, *Rhus chinensis* Mill. (heliophila), and *Myrsine africana* L. and *Vitex negundo* L. (shade-tolerant species) (Table S1), and the herbs were mainly *Carex tristachya* Thunb., *Oplismenus compositus* (L.) Beauv., and *Adiantum capillus-veneris* L.

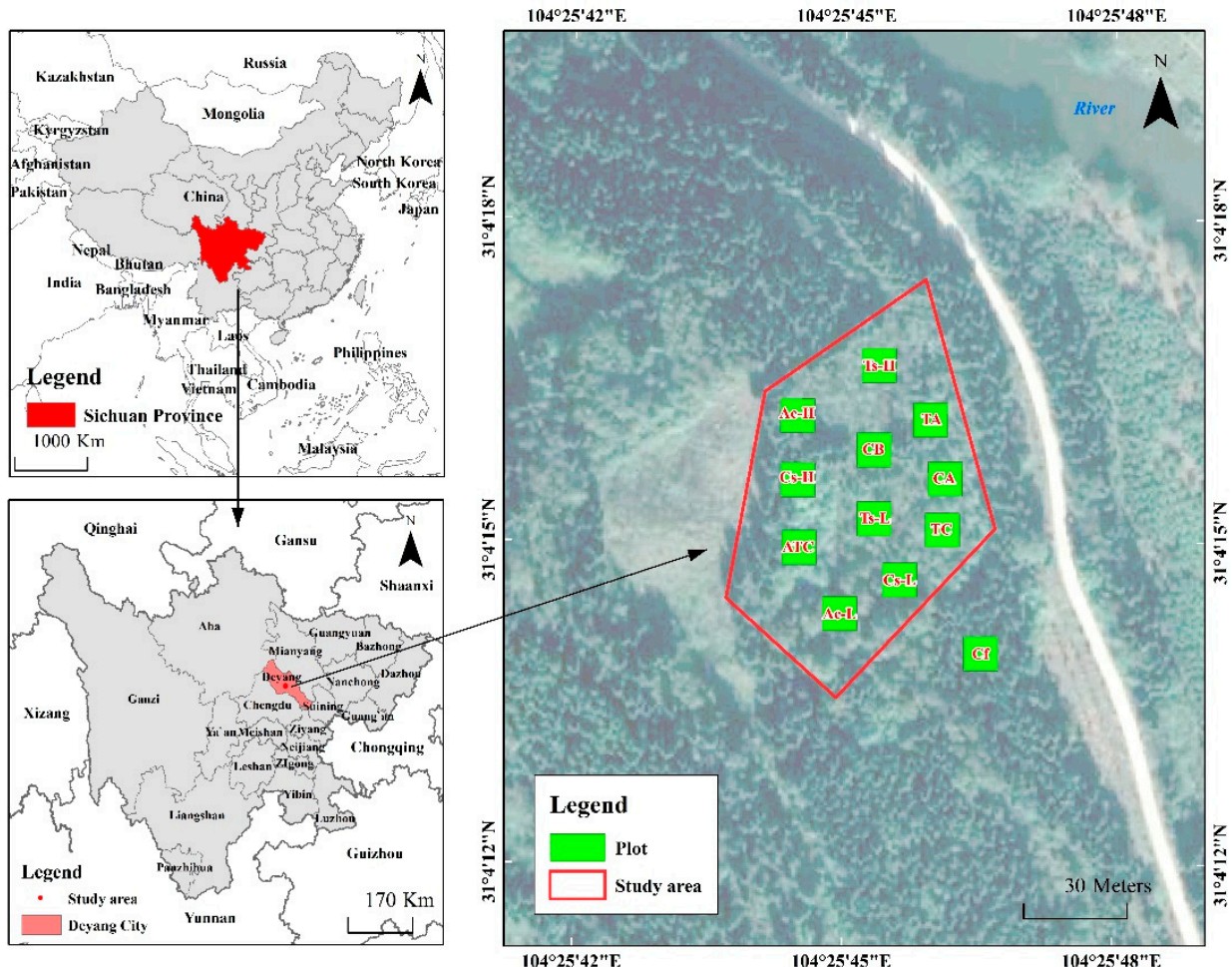

**Figure 1.** Geographical location of the study area.

In 2011, we cut all trees and created different reconstruction patterns for stands in 11 plots to manipulate the canopy composition and density. More specifically, *A. cremastogyne* (Ac-H), *T. sinensis* (Ts-H), and *C. septentrionale* (Cs-H) with a high density of 1.5 m × 2 m; *A. cremastogyne* (Ac-L); *T. sinensis* (Ts-L) and *C. septentrionale* (Cs-L) with a low density of

1.5 m × 4 m; mixed forest patterns of *T. sinensis* + *C. septentrionale* (TC), *C. septentrionale* + *A. cremastogyne* (CA), *T. sinensis* + *A. cremastogyne* (TA), and *A. cremastogyne* + *T. sinensis* + *C. septentrionale* (ATC) with a density of 1.5 m × 2 m; as well as the cutting-blank were applied in the *C. funebris* plantations. A pure *C. funebris* forest was set as the control. The sample plot with a height × width of 10 m × 10 m was chosen for each reconstruction pattern and repeated three times. The basic information of the experimental site in August 2016 is shown in Table 1.

**Table 1.** Vegetation characteristics of experimental plots.

| Pattern | Main Arbor Species | Average Height/m | Average DBH/cm | Canopy Density | Slope/° | Aspect |
|---|---|---|---|---|---|---|
| Ac-H | *A. cremastogyne* | 5.3 | 3.5 | 0.7 | 25° | NE68° |
| Ts-H | *T. sinensis* | 4.1 | 4.3 | 0.6 | 26° | NE68° |
| Cs-H | *C. septentrionale* | 4.2 | 3.4 | 0.6 | 27° | NE68° |
| TC | *T. sinensis* | 4.4 | 3.7 | 0.7 | 27° | NE68° |
| | *C. septentrionale* | 5.3 | 4.5 | | | |
| CA | *C. septentrionale* | 5.7 | 4.7 | 0.7 | 26° | NE68° |
| | *A. cremastogyne* | 5.4 | 5.0 | | | |
| TA | *A. cremastogyne* | 6.1 | 4.9 | 0.7 | 24° | NE68° |
| | *T. sinensis* | 3.8 | 3.6 | | | |
| ATC | *A. cremastogyne* | 5.3 | 4.5 | 0.7 | 23° | SE15° |
| | *T. sinensis* | 4.8 | 4 | | | |
| | *C. septentrionale* | 4.7 | 3.8 | | | |
| Ac-L | *A. cremastogyne* | 6 | 5.3 | 0.6 | 21° | SE24° |
| Ts-L | *T. sinensis* | 4.5 | 3.7 | 0.5 | 24° | SE22° |
| Cs-L | *C. septentrionale* | 4.4 | 4.2 | 0.5 | 22° | SE28° |
| CB | cutting-blank | 4.9 | 4.2 | 0.5 | 25° | NE68° |
| Cf | *Cupressus funebris* | 9.2 | 8.8 | 0.8 | 24° | E |

Ac-H: high-density Alnus cremastogyne, Ts-H: high-density Toona sinensis, Cs-H: high-density Cinnamomum septentrionale, TC: Toona sinensis + Cinnamomum septentrionale, CA: Cinnamomum septentrionale + Alnus cremastogyne, TA: Toona sinensis + Alnus cremastogyne, ATC: Alnus cremastogyne + Toona sinensis + Cinnamomum septentrionale, Ac-L: low-density Alnus cremastogyne, Ts-L: low-density Toona sinensis, Cs-L: low-density Cinnamomum septentrionale, CB: cutting-blank, Cf: Cupressus funebris. High density: 1.5 m × 2 m; low density: 1.5 m × 4 m.

### 2.2. Survey Methods

Three samples (10 m × 10 m) for each pattern were set in August 2016, and four 5 m × 5 m fixed small plots were arranged at the four corners of the samples to investigate the shrub plants. The species, quantity, average height, and coverage of shrub plants were recorded in the plots. Due to the high number of all shrub species and the low frequency of occurrence of some species, only 16 dominant species that appeared in 3 or more sample plots were studied in terms of their ecological niche characteristics.

### 2.3. Measurement Methods

(1) Diversity

We adopted the following α diversity correlation index in this study [23,24]:

(a) The Shannon–Wiener (*H*) and Simpson (*H′*) indexes were used for the calculation of the diversity index:

$$H = -\sum_{i=1}^{s} p_i \times \log_{pi} \tag{1}$$

$$H' = 1 - \sum_{i=1}^{s} p_i^2 \tag{2}$$

(b) The Pielou index was used as the evenness index:

$$J_{sw} = \frac{-\sum_{i=1}^{s} P_i \ln P_i}{\ln S} \tag{3}$$

(c)  Margalef richness index:

$$R = (S - 1) / \ln(N) \tag{4}$$

where $S$ is the number of species in the sample; $N$ is the total quantity of species; $p_i = n_i / N$; and $n_i$ is the number of species $i$.

(2)  Importance values = (relative density + relative frequency + relative coverage)/3
(3)  Niche breadth

The Shannon–Wiener niche breadth index was adopted to indicate the niche breadth [25]:

$$B_i = - \sum_{j=1}^{r} P_{ij} \ln P_{ij} \tag{5}$$

where $B_i$ is the niche breadth of species $i$; $P_{ij} = n_{ij}/N_i$, $n_{ij}$ is the importance value of species $i$ for resource $j$; and $N_i$ is the sum of the importance values of species $i$ for all resources. $P_{ij}$ represents the proportion of the importance value of species $i$ for resource $j$ to that of all resources; $r$ is the sum of the resource niche; and $j$ represents the different reconstruction patterns.

(4)  Niche overlap

The Levins [26] niche overlap expression is as follows:

$$L_{ih} = B_{(L)i} \sum_{j=1}^{r} P_{ij} \times P_{hj} \tag{6}$$

$$L_{hi} = B_{(L)h} \sum_{j=1}^{r} P_{ij} \times P_{hj} \tag{7}$$

$$B_{(L)i} = 1 / \left( r \sum_{j=1}^{r} P_{ij}^2 \right) \tag{8}$$

where $L_{ih}$ is the niche overlap of species $i$ on species $h$; $L_{hi}$ is the niche overlap of species $h$ on species $i$; $P_{ij} = n_{ij}/N_i$; and $N_i = \sum_{j=1}^{r} n_{ij}$; $n_{ij}$ is the species' importance value; $B_{(L)}$ is the Levins niche breadth index; $B_{(L)I}$ and $B_{(L)h}$ have the range [1/r,1]; and $L_{ih}$ and $L_{hi}$ have the range [0, 1].

(5)  Canopy closure

In each plot, canopy closure was estimated with a gridded concave mirror (spherical crown densitometer) in each of the four azimuth directions at a central point offset by at least 2 m from the nearest tree. Briefly, the grid on the mirror was used to count points at crossing lines that coincided with the tree canopy on the mirror, calculated as the percentage of canopy closure.

### 2.4. Data Processing and Statistical Analysis

SPSS software 19.0 (SPSS Inc., Chicago, IL, USA) was used to analyze the experimental data. Generalized linear models (GLMs) were used to analyze the linear distribution and identity link functions, and one-way ANOVA was used to analyze the diversity of shrubs under different reconstruction patterns, followed by Fisher's least significant differences (LSD) post hoc test. GraphPad Prism 5 (GraphPad Software Inc., San Diego, CA, USA) was used for graph drawing.

## 3. Results

### 3.1. Shrub Diversity among Different Overstory Densities

In the pure forest structure, although the Simpson diversity index and Pielou evenness index did not improve significantly ($p < 0.05$), the Shannon–Wiener diversity index and

the Margalef richness index of understory shrubs increased significantly (Figure 2; GLMs, $p < 0.05$). Therefore, overall, in addition to the evenness index, the diversity of understory shrubs in the six patterns of *A. cremastogyne* (A), *T. sinensis* (B), and *C. septentrionale* (C) as a broad-leaved forest is higher than that of a pure *C. funebris* forest as a coniferous forest. For stands of different densities, although the differences in the indices of the high and low densities of the Ac forest are not significant, for the Ts and Cs, the diversity of understory shrubs with high density is better than that of pure low-density forests. In general, in addition to the Pielou evenness index, the shrub diversity of the modified broad-leaved forest model is higher than that of the pure *C. funebris* forest, and the improvement degree of the high-density model is relatively higher than that of the low-density patterns.

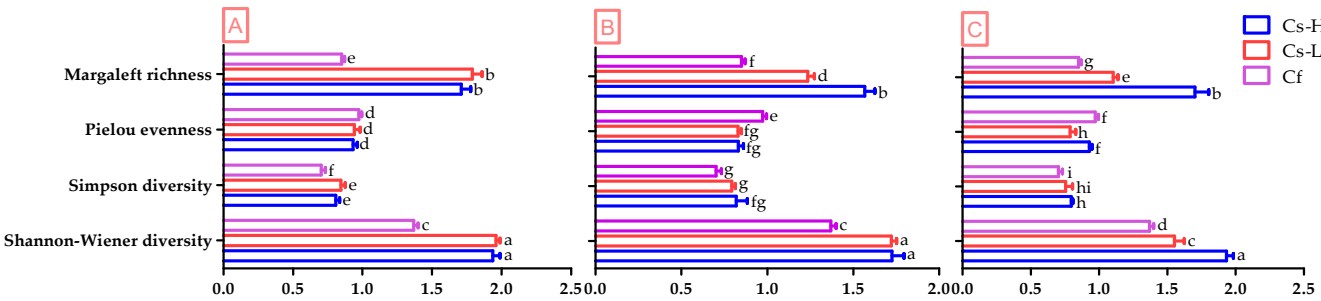

**Figure 2.** Shrub diversity in *Cupressus funebris* stands under pure forest reconstruction patterns at different densities. Shrub diversities (mean ± SD, $n = 3$) in *Cupressus funebris* after reconstructions of *Alnus cremastogyne* (**A**), *Toona sinensis* (**B**), and *Cinnamomum septentrionale* (**C**) at high or low intensity were investigated. Different letters indicate significant differences for the same diversity index among different intensities within the same reconstruction patterns compared by Fisher's least significant differences (LSD) test at $p < 0.05$.

This study included a diversity gradient from monocultures to three species mixtures, and in the two mixed cultivation model forests, three tree species collocations were included. Compared with Cf, the diversity of understory shrubs was significantly affected by the transformation in all four indexes. Specifically, the Shannon–Wiener diversity index and Margalef richness index of shrubs under reconstruction models of high-density pure forests and mixed forests were significantly higher than those of Cf (Figure 3A,D; one-way ANOVA, $p < 0.05$). The overall CA, TA, and TC are higher than those of the pure forest model of a single tree species and the model of mixed cultivation of three tree species. Except for ATC, the Simpson diversity index of shrubs under the reconstruction model of a high-density pure forest and mixed forest was significantly higher than that of Cf (Figure 3B; one-way ANOVA, $p < 0.05$), but there is no significant difference between the reconstruction patterns of different mixed-diversity gradients. Regarding the Pielou evenness index, the overall transformation patterns of high-density pure forests and mixed forests are significantly lower than those of untreated Cf (Figure 3C, one-way ANOVA, $p < 0.05$). In short, for different gradients of mixed diversity, the shrub diversity improvement effect under the two-tree mixed cultivation patterns is the best, followed by the pure forest with a single tree species and, finally, the mixed forest with three tree species and mixed cultivation. However, in the mixed cultivation patterns of two tree species, there is no obvious difference in the diversity of shrubs among the mixed modes of different collocations.

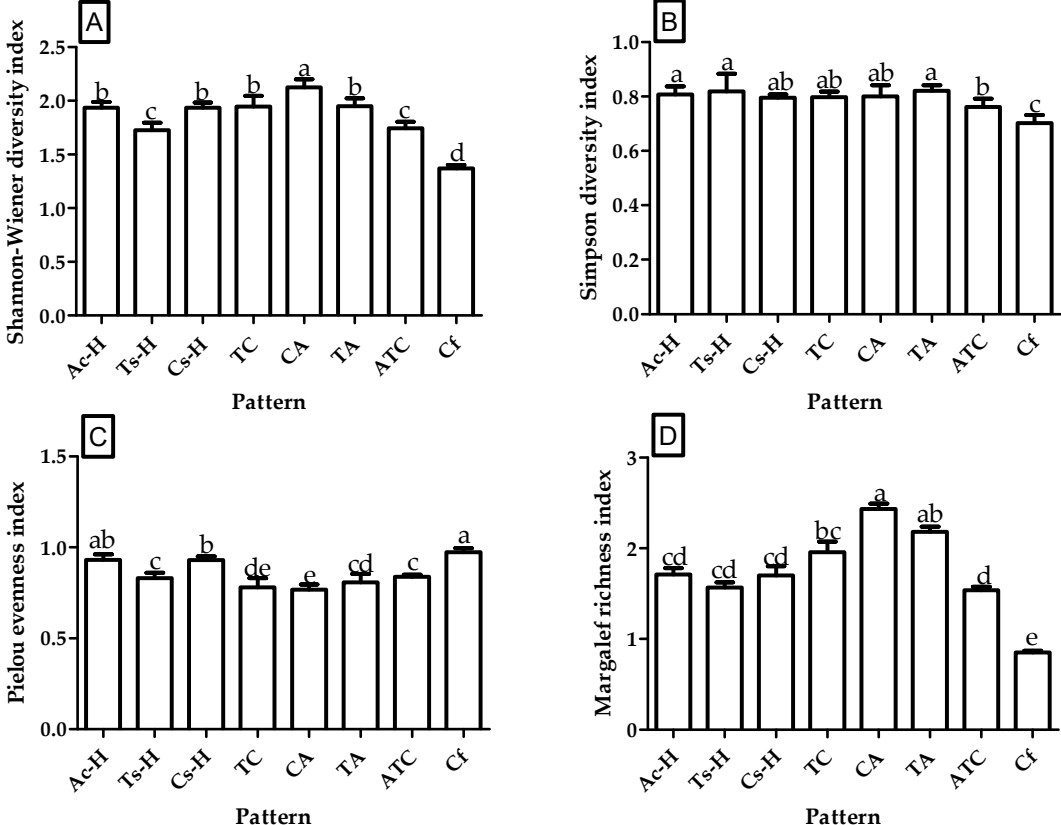

**Figure 3.** The diversity of shrubs among the different tree species component reconstruction patterns of low-efficiency stands of *Cupressus funebris*. The (**A**) Shannon–Wiener diversity, (**B**) Simpson diversity, (**C**) Pielou evenness, and (**D**) Margalef richness indices of shrubs under different reconstruction patterns were investigated. Each bar denotes three replicates (±SD). Different letters indicate significant differences among different reconstruction patterns compared by Fisher's least significant differences (LSD) test at *p* < 0.05. Ac-H: high-density *Alnus cremastogyne*; Ts-H: high-density *Toona sinensis*; Cs-H: high-density *Cinnamomum septentrionale*; TC: Ts + Cs; CA: Cs + Ac; TA: Ts + Ac; ATC: Ac + Ts+ Cs; CB: cutting-blank; Cf: *Cupressus funebris.* High density: 1.5 m × 2 m; low density: 1.5 m × 4 m.

### 3.2. Shrub Niche Breadth

As we can see from Table 2, the total niche breadths of *B. papyrifera* (2.371), *R. chinensis* (2.296), *Vitex negundo* (2.191), and *Zanthoxylum bungeanum* Maxim (2.092) were higher than the rest of the shrubs, showing that they had a wider range of environmental resource utilization and a dominant position in the shrub population. The niche breadth of *B. papyrifera* was the highest in the Ts-H, Ts-L, Cs-H, Cs-L, ATC, and CB patterns; the niche breadth of *V. negundo* was the highest in the TC and Ac-L patterns, while the niche breaths of *Pyracantha fortuneana* (Maxim.) Li, *Elaeagnus pungens* Thunb, and *R. chinensis* were the highest in the TA, Ac-H, Cf, and CA patterns, respectively. It can be seen that *B. papyrifera*, a heliophyte, occupied a dominant position in these patterns with relatively low canopy density. The shade-tolerant species, such as *V. negundo* and *M. africana*, began to occupy a dominant position with the gradually increasing canopy closure. However, as all reconstruction patterns were still in the initial stage, the canopy density was relatively low. *B. papyrifera* occupied a higher ecological status in all reconstruction patterns, except for the unreconstructed Cf pattern.

### 3.3. Shrub Importance Values and Niche Overlap

It can be seen from Table 3 that the shrubs occupying a dominant position in the stand under different transformation modes are different, especially regarding their density. For example, in lower-density models, such as Ts-L, Cs-L, and CB, *B. papyrifera*, *R. chinensis*, and

*V. negundo* are dominant, while in Ac-H, Ts-H, Cs-H, TC, CA In, TA, and ATC, although *B. papyrifera* also occupies a dominant position in some models, there are also some shrubs that do not occupy a dominant position in lower-density models, such as *M. africana*, *Vernicia fordii* (Hemsl.) Airy Shaw and *A. chinense*. The species with a relatively large total shrub importance value are *B. papyrifera*, *R. chinensis*, and V. negundo, which shows that they are more adaptable to the environment and make fuller use of environmental resources.

**Table 2.** The niche breadths of dominant shrub populations under different reconstruction patterns.

| Species | Total | Niche Breadth | | | | | | | | | | | |
| | | Pattern | | | | | | | | | | | |
| | | Ac-H | Ts-H | Cs-H | TC | CA | TA | ATC | Ac-L | Ts-L | Cs-L | CB | Cf |
| *Alangium chinense* | 2.057 | | | | 1.075 | | | | 1.087 | 0.693 | | 0.691 | 0.690 |
| *Cupressus funebris* | 1.839 | 1.087 | | | | 1.090 | 1.048 | 0.614 | | | | | 0.671 |
| *Rubus coreanus* | 1.886 | 0.681 | 1.074 | 0.645 | | 0.927 | 0.958 | 0.693 | 0.690 | | | | |
| *Broussonetia papyrifera* | 2.371 | 1.337 | 1.373 | 1.373 | 1.090 | 1.030 | 1.035 | 1.351 | 1.336 | 1.383 | 1.369 | 1.371 | |
| *Toona ciliata* | 1.075 | | | | | | | | | | | | |
| *Elaeagnus pungens* | 1.805 | 1.371 | 1.051 | | | | | 1.024 | 0.660 | | | | |
| *Zanthoxylum bungeanum* | 2.092 | 1.098 | 0.583 | | 0.693 | 0.691 | | 1.088 | 1.349 | 1.330 | 1.085 | | |
| *Vitex negundo* | 2.191 | 0.647 | | 1.065 | 1.365 | 1.003 | 1.070 | 1.340 | 1.374 | | 1.306 | 1.092 | 1.084 |
| *Pyracantha fortuneana* | 2.027 | | 1.045 | | 0.581 | 0.621 | 1.350 | 0.675 | 0.631 | 0.688 | | 1.098 | |
| *Coriaria nepalensis* | 1.537 | | | | | | 0.686 | | | | | | |
| *Ligustrum lucidum* | 1.751 | 1.368 | 0.636 | 1.080 | | 0.675 | | | | | | | |
| *Rhamnus davurica* | 1.591 | | | | | | | | | | | | |
| *Myrsine africana* | 2.028 | 0.665 | 1.343 | 1.095 | | 0.660 | | 0.692 | 0.655 | 0.681 | | 0.693 | 1.357 |
| *Sapium sebiferum* | 1.560 | 0.689 | | | | | 0.685 | | | | | | |
| *Rhus chinensis* | 2.296 | 0.676 | 1.286 | 1.359 | 1.008 | 1.381 | 1.101 | 1.024 | 1.093 | 1.379 | 1.336 | 1.098 | |
| *Vernicia fordii* | 1.771 | 0.652 | 1.351 | 0.658 | | | | | 0.684 | | | 0.684 | |

Ac-H: high-density *Alnus cremastogyne*; Ts-H: high-density *Toona sinensis*; Cs-H: high-density *Cinnamomum septentrionale*; TC: Ts + Cs; CA: Cs + Ac; TA: Ts + Ac; ATC: Ac + Ts+ Cs; Ac-L: low-density *Alnus cremastogyne*; Ts-L: low-density *Toona sinensis*; Cs-L: low-density *Cinnamomum septentrionale*; CB: cutting-blank; Cf: *Cupressus funebris.* High density: 1.5 m × 2 m; low density: 1.5 m × 4 m.

**Table 3.** The importance values of dominant shrub populations under different reconstruction patterns.

| Species | Total | Importance Values | | | | | | | | | | | |
| | | Pattern | | | | | | | | | | | |
| | | Ac-H | Ts-H | Cs-H | TC | CA | TA | ATC | AcII | TsII | CsII | CB | Cf |
| *Alangium chinense* | 4.856 | 1.512 | 1.497 | 2.954 | 5.712 | | 1.582 | 1.888 | 11.490 | 6.478 | 2.762 | 10.388 | 17.633 |
| *Cupressus funebris* | 3.264 | 7.823 | 3.263 | 1.989 | 1.308 | 7.819 | 9.591 | 3.377 | | | | | 15.016 |
| *Rubus coreanus* | 5.151 | 6.324 | 8.499 | 6.602 | 1.578 | 21.699 | 14.215 | 2.766 | 2.960 | | | 2.515 | |
| *Broussonetia papyrifera* | 18.160 | 23.396 | 10.650 | 20.970 | 11.953 | 10.909 | 9.422 | 22.803 | 14.668 | 26.653 | 32.873 | 31.286 | 5.729 |
| *Toona ciliata* | 0.827 | | 1.497 | | | 2.491 | 1.768 | | | | | | |
| *Elaeagnus pungens* | 2.891 | 8.100 | 8.494 | 1.798 | | 2.727 | | 5.012 | 4.117 | 2.462 | | | |
| *Zanthoxylum bungeanum* | 7.397 | 8.196 | 4.961 | | 3.333 | 3.948 | 1.582 | 7.543 | 12.928 | 20.633 | 13.823 | | 6.514 |
| *Vitex negundo* | 11.878 | 4.749 | | 7.394 | 28.201 | 13.467 | 13.774 | 21.755 | 14.121 | | 20.297 | 11.352 | 27.113 |
| *Pyracantha fortuneana* | 5.730 | 3.110 | 6.478 | | 5.533 | 6.840 | 20.598 | 3.525 | 6.140 | 7.555 | | 12.201 | |
| *Coriaria nepalensis* | 1.507 | | 1.370 | | | 3.489 | 3.529 | 1.517 | | 2.484 | | | |
| *Ligustrum lucidum* | 2.522 | 8.668 | 3.151 | 8.223 | 1.494 | 4.446 | 2.718 | 1.651 | | | | | |
| *Rhamnus davurica* | 1.400 | 1.646 | 1.795 | 2.409 | | 2.491 | 1.610 | | | | | | |
| *Myrsine africana* | 5.460 | 7.813 | 16.688 | 8.242 | 3.793 | 4.293 | | 2.567 | 4.429 | 5.734 | | 7.388 | 27.993 |
| *Sapium sebiferum* | 1.320 | 3.138 | | | | 1.757 | 2.703 | 1.200 | | | 2.411 | | |
| *Rhus chinensis* | 17.036 | 4.728 | 19.000 | 28.872 | 14.052 | 11.543 | 8.248 | 20.247 | 16.611 | 28.000 | 27.833 | 16.092 | |
| *Vernicia fordii* | 2.736 | 7.890 | 6.941 | 3.973 | 1.865 | | 1.410 | | 2.933 | | | 8.776 | |

Ac-H: high-density *Alnus cremastogyne*; Ts-H: high-density *Toona sinensis*; Cs-H: high-density *Cinnamomum septentrionale*; TC: Ts + Cs; CA: Cs + Ac; TA: Ts + Ac; ATC: Ac + Ts+ Cs; Ac-L: low-density *Alnus cremastogyne*; Ts-L: low-density *Toona sinensis*; Cs-L: low-density *Cinnamomum septentrionale*; CB: cutting-blank; Cf: *Cupressus funebri.* High density: 1.5 m × 2 m; low density: 1.5 m × 4 m.

According to Tables 3 and S2, the niche overlap of dominant shrubs in the patterns of different compositions and densities can be obtained (Table 4), and the diversity of shrubs is greater in the patterns with lower niche overlap of dominant shrubs. For example, the

overlap of dominant shrubs in high-density pure forests is lower than that of low-density pure forests, and the overlaps of dominant shrubs in the Cf pattern are the highest, while the overlap of dominant shrubs in the two mixed tree species patterns is lower than that of pure forests and the three tree species mixed with the same density, which is opposite to the distribution of shrub diversity in each model.

**Table 4.** The niche overlap of dominant shrubs in different reconstruction patterns.

| Patterns | Dominant Shrub | Importance Value | Proportion of Total Importance Value (%) | Niche Overlap | Average Niche Overlap |
|---|---|---|---|---|---|
| Cf | *Myrsine africana* | 27.993 | | 0.592 | |
| | *Vitex negundo* | 27.113 | 72.739 | 0.799 | 0.7103 ± 0.1066 |
| | *Alangium chinense* | 17.633 | | 0.74 | |
| Ts-H | *Myrsine africana* | 16.688 | | 0.469 | |
| | *Broussonetia papyrifera* | 10.65 | 35.837 | 0.918 | 0.6293 ± 0.2505 |
| | *Rubus coreanus* | 8.499 | | 0.501 | |
| Cs-H | *Rhus chinensis* | 28.872 | | 0.918 | |
| | *Broussonetia papyrifera* | 20.97 | 58.084 | 0.469 | 0.6293 ± 0.2505 |
| | *Myrsine africana* | 8.242 | | 0.501 | |
| Ac-H | *Broussonetia papyrifera* | 23.396 | | 0.561 | |
| | *Ligustrum lucidum* | 8.668 | 40.26 | 0.78 | 0.5443 ± 0.2444 |
| | *Zanthoxylum bungeanum* | 8.196 | | 0.292 | |
| Ts-L | *Rhus chinensis* | 28 | | 0.918 | |
| | *Broussonetia papyrifera* | 26.653 | 75.286 | 0.771 | 0.8230 ± 0.0824 |
| | *Zanthoxylum bungeanum* | 20.633 | | 0.78 | |
| Cs-L | *Broussonetia papyrifera* | 32.873 | | 0.981 | |
| | *Rhus chinensis* | 27.833 | 81.003 | 0.692 | 0.7713 ± 0.1834 |
| | *Vitex negundo* | 20.297 | | 0.641 | |
| Ac-L | *Rhus chinensis* | 16.611 | | 0.918 | |
| | *Broussonetia papyrifera* | 14.668 | 45.4 | 0.641 | 0.7797 ± 0.1385 |
| | *Vitex negundo* | 14.121 | | 0.78 | |
| TA | *Pyracantha fortuneana* | 20.598 | | 0.702 | |
| | *Rubus coreanus* | 14.215 | 48.587 | 0.528 | 0.5597 ± 0.1294 |
| | *Vitex negundo* | 13.774 | | 0.449 | |
| TC | *Vitex negundo* | 28.201 | | 0.641 | |
| | *Rhus chinensis* | 14.052 | 54.206 | 0.692 | 0.7503 ± 0.1474 |
| | *Broussonetia papyrifera* | 11.953 | | 0.918 | |
| CA | *Rubus coreanus* | 21.699 | | 0.449 | |
| | *Vitex negundo* | 13.467 | 46.709 | 0.489 | 0.5263 ± 0.1013 |
| | *Rhus chinensis* | 11.543 | | 0.641 | |
| ATC | *Broussonetia papyrifera* | 22.803 | | 0.692 | |
| | *Vitex negundo* | 21.755 | 64.805 | 0.918 | 0.7503 ± 0.1474 |
| | *Rhus chinensis* | 20.247 | | 0.641 | |
| CB | *Broussonetia papyrifera* | 31.286 | | 0.918 | |
| | *Rhus chinensis* | 16.092 | 59.579 | 0.612 | 0.7010 ± 0.1889 |
| | *Pyracantha fortuneana* | 12.201 | | 0.573 | |

Ac-H: high-density *Alnus cremastogyne*; Ts-H: high-density *Toona sinensis*; Cs-H: high-density *Cinnamomum septentrionale*; TC: Ts + Cs; CA: Cs + Ac; TA: Ts + Ac; ATC: Ac + Ts + Cs; Ac-L: low-density *Alnus cremastogyne*; Ts-L: low-density *Toona sinensis*; Cs-L: low-density *Cinnamomum septentrionale*; CB: cutting-blank; Cf: *Cupressus funebri*. High density: 1.5 m × 2 m; low density: 1.5 m × 4 m.

## 4. Discussion

For plantations with poor tree diversity, understory vegetation is an important part of the diversity of the entire plantation, providing a habitat and food for many organisms, promoting nutrient cycling, preventing erosion, and affecting seed germination and

seedlings in the forest ecosystem with regard to survival and growth [22]. In this study, we carried out different patterns of cypress forest transformation, and studied and revealed the diversity of understory shrubs in stands with different tree compositions and density and the relationship between their shrub niches. The results showed that the tree composition and density had a significant impact on the diversity of shrubs, and that the niche is closely related to the diversity of shrubs. In each transformation pattern, the overlap of the niche of the shrub diversity and the dominant shrub had an opposite trend.

There are two hypotheses that can be used to explain plant diversity: resource heterogeneity and resource quantity [27,28]. First, the resource heterogeneity hypothesis states that plants have different adaptabilities to different environmental resources; therefore, plant diversity will change with differences in habitats or resource heterogeneity [27,28]. Environmental heterogeneity is more widely recognized to enhance biodiversity than homogenous environments through greater niche space [4], while the resource quantity hypothesis holds that the limitation of resource quantity is the main reason for maintaining plant diversity [29,30]. Indeed, the more-individuals hypothesis [31] is an extension of the resource quantity hypothesis. Srivastava and Lawton [31] believe that higher energy availability promotes a higher number of individuals in a community, which consequently increases species richness and species diversity. However, Storch David [32] believes that maintaining a higher number of individuals with more energy is only part of the resource quantity hypothesis on diversity, in which resources can not only increase the number of individuals to improve species richness, but also improve species richness by influencing diversity patterns in space and time. In brief, resource quantity is undoubtedly an important factor in species diversity.

Studies have shown that the understory environment is variable in space and time, which mainly depends on the composition of tree species, stand density, stand structure, and canopy pattern, including the spatial arrangement of the canopy and canopy gaps [33]. Planting density is regarded as an important factor affecting the diversity of understory shrubs. Planting density will affect the dominant ecological factors in the growth and development of understory shrubs, such as light and moisture [34,35]. In the patterns of different densities after this transformation, the change in planting density due to canopy closure is an important factor affecting the diversity of shrubs. In addition to the Pielou evenness index, the high-density pure stands have higher shrub diversity than the low-density pure stands, but the densest pure *C. funebris* forest had lower shrub diversity (Figure S1). This may be due to the fact that when the density is low, the canopy closure is low, and the light resources are sufficient. The heliophila, *B. papyrifera* and *R. chinensis*, possessing strong competitiveness, grew fast, and occupied most of the light resources under the forest, resulting in low shrub diversity. However, the canopy of the tree layer brought more shade to the understory with the increase in canopy closure, which increased the degree of heterogeneity of light resources and provided more niche space. Thus, some shade-tolerant species, such as *V. negundo* and *M. africana*, appeared in the stand and consequently prompted greater plant diversity. This is also observed in the low-density model, where *B. papyrifera* and *R. chinensis* occupy a larger niche width, and in the high-density model, in which the niches of *V. negundo* and *M. Africana*—the shade-tolerant plants—gradually increase. For example, in the study of Putten et al., a phenomenon occurred in which lower light levels in high-density forest stands were beneficial to shade-loving plants below the canopy [36,37]. However, the diversity decreased gradually with the further increase in canopy closure. Although this might result from many factors, the light resource shortage in this situation, which is insufficient to support the growth of more shrubs, could be another cause. In this study, the diversity of understory shrubs was the comprehensive embodiment of resource heterogeneity and resource quantity. The reason that the shrub evenness under each reconstruction pattern was lower than that of the unreconstructed pure *C. funebris* forest may be that the undergrowth plant species of the pure *C. funebris* forest were scarce, the distribution was sparse, and the community was

also in the stable stage of the late succession phase; therefore, the shrub distribution under the pure *C. funebris* forest was more even than that of other patterns.

At the same time, the overstory composition changed from the original pure *C. funebris* forest to a mixed forest with different combinations of Ac, Ts, and Cs, which are deciduous broad-leaved trees. Therefore, a transformation occurred from an evergreen coniferous forest to a deciduous broad-leaved forest, which can increase litter production and is favorable for the growth of soil microorganisms [38]. The ecological process in the stand became more complex after the reconstruction from a pure C. *funebris* forest to a mixed forest, where more diversified stands led to greater environmental heterogeneity in general [7]. The mixed forest differed from the pure forest not only in the spatial aspect, but also in the temporal aspect, as the light changes greatly in different seasons due to the different phenology of species in a mixed forest than that of a pure forest [33,39]. Some studies have found that the diversity of understory plants is higher in a mixed forest than in a pure forest [40], which is consistent with our study. However, in contrast to the prediction, the diversity of shrubs under the ATC with a greater degree of mixing is lower than that of the mixed patterns of the two tree species, which is consistent with the results of Ampoorter, E. et al. [40]. This may be due to the complementary canopy structure and plasticity of mixed forests [33], which makes the stand productivity of mixed forests generally higher than that of pure forests, while also intercepting more light resources, which is not conducive to the improvement of shrub plant diversity [33]. The difference in shrub diversity between mixed forests composed of different tree species is not obvious. This may be due to the fact that the transformed plantation is in the early stage of succession, and the interaction between trees has not been fully manifested.

Niche overlap is generally used to refer to the level of demand for the same resource for two or more species, so niche overlap tends to be larger for species with similar biotypes [10,11]. The greater the degree of niche overlap, the lower the biodiversity [41]. According to the hypothesis of Gause [42], if the niche overlap of two species occurs, it will inevitably lead to competition, which means that the greater the niche overlap between species, the more intense the competition between them. Therefore, niche differentiation is widely regarded as the main driving force for species coexistence and helps maintain biodiversity [10,11]. In this study, the niche overlap between the dominant shrubs in the Cs-H, Ts-H, and Ac-H patterns is less than that in Cs-L, Ts-L, Ac-L, and Cf. This shows that the high degree of niche differentiation in the high-density model is more conducive to the generation of shrub diversity. The lower the degree of niche overlap, the more favorable the symbiosis of species, which is in accordance with the findings of Wittman et al. [43]. Similarly, the result that the niche overlap in the diversity of different mixed tree species and the diversity of shrubs show the opposite trend can also be reflected. The niche overlap between the shrubs occupying a dominant position in the CA, TC, and AC models is generally lower than that of the pure forest and ATC models. Among them, the ATC model has the largest niche overlap compared with the transformed pure forest and the two species. In mixed forests, the diversity of shrubs in the ATC model is relatively poor. Although *B. papyrifera* still occupies a dominant position in high-density forest stands compared with low-density forest stands, there are more shade-tolerant tree species in high-density forest stands. We speculate that this is due to the fact that the transformation is still in the early stage, and the advantages of shade tolerance have not yet been fully realized. In the future, the composition and structure of the arbor canopy may tend to be stable, and the relationship between species may change in the development and succession of the community. Shade-tolerant species will replace *B. papyrifera* and will occupy a more dominant position.

Our research provides ideas for the transformation of low-efficiency plantations in the upper reaches of the Yangtze River and, at the same time, contributes to a deeper understanding of the relationship between the diversity of overstory and shrubs, as well as linking shrub diversity and niche characteristics, which is important for niche theory in the formation and maintenance of diversity. However, this study only shows that the



ecological benefits of planted forests can be improved with suitable tree composition and structure, while the relative importance of the impact of different factors on the diversity of shrubs needs to be explored further. Moreover, the experimental investigation was carried out in the early stage of transformation when the interrelationship between trees may not be very close. Therefore, the difference in the influence of different tree compositions on understory shrubs in mixed forests was not very obvious, and understory shrubs were still in the process of gradually tending toward stable development.

## 5. Conclusions

In the early stage of the transformation, the diversity of shrubs under different patterns showed great differences. Among them, the transformation of the TA, CA, and TC patterns was the most conducive to the improvement of shrub diversity. The different compositions and densities of trees have a significant impact on the diversity of shrubs. In this study, the planting model of broad-leaved trees instead of conifers was more conducive to improving the diversity of shrubs; the shrub diversity first increased and then decreased with the increase in density. The trend of increasing and then decreasing was the same under different mixed tree species diversity gradients, but there was no significant difference in shrub diversity in the mixed mode of different tree species. In the patterns where the niche overlap between the dominant shrubs was low, the diversity of shrubs was greater, which further proves that niche and diversity are closely related, and that the degree of niche overlap can effectively predict diversity.

**Supplementary Materials:** The following are available online at https://www.mdpi.com/article/10.3390/f12111492/s1, Figure S1: Diversity index at different canopy closures after forest reconstruction (all patterns). Table S1: Shrubs and their basic growth characteristics. Table S2: The niche overlap of shrub populations under different reconstruction patterns.

**Author Contributions:** Conceptualization, Y.W. (Yang Wang), S.C., X.L. and C.F.; Formal analysis, Y.W. (Yang Wang) and G.C.; Funding acquisition, X.L. and C.F.; Investigation, S.C., W.H., J.R., X.W., Y.W. (Yu Wang) and M.F.; Methodology, X.L., G.C. and C.F.; Project administration, X.L. and C.F.; Writing—original draft, S.C. and C.F.; Writing—review and editing, Y.W. (Yang Wang) and C.F. All authors have read and agreed to the published version of the manuscript.

**Funding:** This study was supported by the German Government loans for Sichuan Forestry Sustainable Management (Grant No. G1403083) and the Key Sci-tech Project of the "12th 5-year Plan" of China (Grant No. 2011BAC09B05).

**Institutional Review Board Statement:** Not applicable.

**Informed Consent Statement:** Not applicable.

**Data Availability Statement:** The datasets generated during and/or analyzed during the current study are available from the corresponding author on reasonable request.

**Acknowledgments:** The authors would like to thank Haifeng Yin, Size Liu, Qian Lv, Bo Li and Chao Luo from the College of Forest, Sichuan Agricultural University, for their participation in data collection.

**Conflicts of Interest:** The authors declare no conflict of interest.

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
