# Peer review of "Shrub Diversity and Niche Characteristics in the Initial Stage of Reconstruction of Low-Efficiency Cupressus funebris Stands"

_forests, doi:10.3390/f12111492_

Round 1

Reviewer 1 Report

Sometimes descriptions of treatments resulted not easy to read. A complementary table after table 1 describing treatments might be useful to clearly present them. Besides, diversity indexes used might be presented in a better way to make lecture easier. In figure 2, when are presented diversity index a, b) and c) it is no easy before reading the foot note what it means.

Is it possible to present in different way tables 2, 3  and 4. It is not easy to follow well the results presented there. There is a lot of information and description for example expressing 24 or 27 pairs, really is not easy for reading.

Shrub species included might be considered pioneer or permanent species, which would be the time for being expected along natural succession. In results when Niche is presented, some species start to be described for example as shade tolerant or in discussion as heliophile  This information might be useful before when species proposed are presented to be used within the treatments.

Also, canopy density used to explain results within the discussion, would require a better description about how as obtained for each plot. Even this variable is associated in discussion with precipitation interception and soil moisture.

Size plot is 10 m x 10 m, is enough for assessing reconstruction? Even the small plots of 5 m x 5 m. If yes, briefly comments might be added to support this decision. The dominant species selected according to the important value had any threshold.

If biodiversity is response variable and treatments can directly influence this value, it is a bias?

Reviewer 2 Report

Although interesting, the study lacks research depth, focusing on the case study in too many unnecessary details, but not on the reasons for which the research was done (i.e., its contribution to the theoretical (conceptual or methodological) advancement of the field, and relevance of the findings to the field. Adding an English level incompatible with the publication of article in an international research journal, the article requires major changes.

The introduction only presents the previous studies, without analyzing them critically in order to emphasize their shortcomings (ambiguities, lacks, uncertainties, errors), at least in the form of a final paragraph placed in the end of the introduction, right before stating the research goals, justifying the need for research. Also, the authors should stress out the significance of their research question, including the novel and original elements.

In the Method section, the authors should keep in their mind that they are not writing a report to the Chinese authorities, but a research article intended for the broad research readership of "Forests". Therefore, Figure 2 should be understood by a Brazilian researcher too; it should not only display the Chinese provinces, and show the map as if China was the only country in the world, but, using the same map-in-map system, show the neighboring countries, allowing the reader to understand the map in an international context.

The discussions are meant to emphasize the importance of the results for the theoretical (conceptual or methodological) advancement of the field by presenting (A) the significance of results - what do they say, in scientific terms; (B) the inner validation of results, against the study goals or hypotheses; (C) the external validation of results, against those of similar studies from other countries, identified in the literature (again, connected to the poor review of the international literature); (D) the importance of the results, meaning their contribution (conceptual or methodological) to the theoretical advancement of the field (related to the lack of a clear theoretical framework); (E) a summary of the study limitations and directions for overcoming them in the future research. Out of these, only sections A through C are present. The authors should also add the missing parts.

The section labeled "Conclusions" does not contain real conclusions, but only a summary of the main findings, pertaining strictly to the case study. Conclusions are meant to deliver a scientific message, far away beyond the case study, to the entire scientific community, making a clear contribution to the theoretical (conceptual or methodological) development of the field; the current conclusions lack research depth, pertaining only to the case study.

Abstract: Please remove the number and headings. According to the Author Guidelines, the abstract is not structured explicitly. Also, the abstract lacks research depth, focusing too much on the case study and on presenting the results, without revealing their broader significance and justifying the need for research and importance of findings.

Language: the level of English is incompatible with the publication of article in an international research journal. Number disagreements (to find a suitable forestry measures), incomprehensible phrasing due to confusions (resource competition did presented, but not intensive) make the manuscript incomprehensible, even starting with the abstract. There are also numerous examples of the wrong use of capitalization, punctuation, spaces etc., harder to indicate provided that the authors altered the journal template, removing the line numbers. The appeal to a native English speaker is an absolute requirement for the publication of manuscript.

Round 2

Reviewer 2 Report

The authors have addressed most comments, but not in sufficient depth, and as a result changes are still necessary to the manuscript before its publication.

The aim of the study is extremely important, and cannot be masked by being part of a larger paragraph. It needs to make a separate paragraph, including a clear statement of the goals (i.e., "this study aims to..." and not like an abstract: "In this study, we selected..."

The abstract is still underdeveloped, and focused on delivering useless figures (four years, 20 and 11 patterns, two densities, 1.5*2 m and 1.5*4 m), instead of ideas. The study goals are missing and the contribution of results to the theoretical advancement of the field insufficiently described.

English was the weakest point, and although the authors claim that "The revised manuscript has been edited by a professional organization", flagrant errors, such as "We are also attempted", are still present. The authors should seek for a professional service, specialized in editing research articles.
